# Role of Polymeric Coating on Metallic Foams to Control the Aeroacoustic Noise Reduction of Airfoils with Permeable Trailing Edges

**DOI:** 10.3390/ma12071087

**Published:** 2019-04-02

**Authors:** Reza Hedayati, Alejandro Rubio Carpio, Salil Luesutthiviboon, Daniele Ragni, Francesco Avallone, Damiano Casalino, Sybrand van der Zwaag

**Affiliations:** 1Novel Aerospace Materials group, Faculty of Aerospace Engineering, Delft University of Technology (TU Delft), Kluyverweg 1, 2629 HS Delft, The Netherlands; s.vanderzwaag@tudelft.nl; 2Section Aircraft Noise & Climate Effects (ANCE), Faculty of Aerospace Engineering, Delft University of Technology (TU Delft), Kluyverweg 1, 2629 HS Delft, The Netherlands; A.RubioCarpio@tudelft.nl (A.R.C.); S.Luesutthiviboon@tudelft.nl (S.L.); 3Aerodynamics, Wind Energy, Flight Performance and Propulsion, Faculty of Aerospace Engineering, Delft University of Technology (TU Delft), Kluyverweg 1, 2629 HS Delft, The Netherlands; d.ragni@tudelft.nl (D.R.); F.Avallone@tudelft.nl (F.A.); d.casalino@tudelft.nl (D.C.)

**Keywords:** aero-acoustics, noise-control, metal foam, trailing edge, spray coating, noise reduction

## Abstract

Studies on porous trailing edges, manufactured with open-cell Ni-Cr-Al foams with sub-millimeter pore sizes, have shown encouraging results for the mitigation of turbulent boundary-layer trailing-edge noise. However, the achieved noise mitigation is typically dependent upon the pore geometry, which is fixed after manufacturing. In this study, a step to control the aeroacoustics effect of such porous trailing edges is taken, by applying a polymeric coating onto the internal foam structure. Using this method, the internal topology of the foam is maintained, but its permeability is significantly affected. This study opens a new possibility of aeroacoustic control, since the polymeric coatings are temperature responsive, and their thickness can be controlled inside the foam. Porous metallic foams with pore sizes of 580, 800, and 1200 μm are (internally) spray-coated with an elastomeric coating. The uncoated and coated foams are characterized in terms of reduced porosity, average coating thickness and air-flow resistance. Subsequently, the coated and uncoated foams are employed to construct tapered inserts installed at the trailing edge of an NACA 0018 airfoil. The noise mitigation performances of the coated metal foams are compared to those of uncoated metal foams with either similar pore size or permeability value, and both are compared to the solid trailing edge reference case. Results show that that the permeability of the foam can be easily altered by the application of an internal coating on the metallic foams. The noise reduction characteristics of the coated foams are similar to equivalent ones with metallic materials, provided that the coating material is rigid enough not to plastically deform under flow conditions.

## 1. Introduction

Aeroacoustic noise pollution has become a major issue that hinders the development of new technologies in several fields such as aerospace, aviation and sustainable energy. For example, the noise produced by airplanes significantly reduces the service hours of airports, which need to comply with noise restrictions in order to operate close to urban areas [1]. In wind energy, wind-turbine noise is one of the major hindrances forcing wind farms to be located far away from highly populated areas, and to limit their power output overnight [2].

It has been shown that in wind turbines, the turbulent boundary-layer trailing-edge (TBL-TE) noise is generally the dominant source of noise [3,4,5]. The main cause of TBL-TE noise is the scattering of turbulent pressure fluctuations at the trailing edge into acoustic pressure waves [6]. While current active-control techniques such as flow suction and blowing have been extended to the noise reduction field [7], many new ones have been inspired by nature, as from an example, those derived by the “silent flight of owls”. Important structural characteristics of owls’ wings, such as the high permeability of their feathers [8], the trailing edge fringes [9], and additional soft upper surfaces of the wings [9,10], have been translated to man-made solutions for airfoils. This has led to the following successful trailing-edge modifications: Serrations [11,12,13], brush extensions [14], and permeable trailing edges [15,16,17]. Measurements performed on airfoils with serration at flow speed of 35 m/s have shown a noise abatement of about 2 dB at 4 kHz, and up to 7 dB at 1 kHz [13,18,19]. Brushed trailing edges with fiber-like structures located at the edge of the blade, which are able to align themselves with the flow [14,20,21], have yielded noise reductions of 2–5 dB over the frequency range of 0.6–2 kHz.

Permeable trailing edges have been investigated in several recent studies [16,22,23,24,25,26,27,28,29]. In one of their first studies on airfoils, Sarradj and Geyer [23] constructed completely porous (polymer, glass or metallic) airfoils of different flow resistivity, and tested their aerodynamic and aeroacoustics performances. Even though their results showed up to 8 dB noise reduction as a result of making the airfoil permeable, the aerodynamic results showed an at least 70% decrease in lift. Moreover, their results did not demonstrate an obvious relation between the material used to construct the foam and the noise reduction, since they had only focused on the effect of the pore size characterized by the flow resistivity. More recently, Geyer and Sarradj [29] investigated partially porous airfoils which were obtained by simply covering part of the frontal surface of their previously tested fully porous airfoils with a thin impermeable foil. They could reach a maximum noise reduction of 10 dB in a Strouhal number range of 10–70. Moreover, increasing the permeability of the porous part increased the boundary layer thickness, the turbulence intensity around the trailing edge and in the wake region, together with the wake deficit.

Herr et al. [28] studied noise reduction in airfoils with porous metal trailing edges covering 10% of chord length. Their partially porous airfoils caused a noise reduction of 2–6 dB as compared to the reference solid airfoils. They also observed that the effect of porous treatment diminishes by increasing the airfoil angle of attack. Rubio Carpio et al. [30] used 10% and 20% chord length Ni-Cr-Al porous inserts with pore sizes of 450, 580, and 800 μm on NACA 0018 airfoils. The results showed that at low frequencies, increasing the angle of attack reduces the noise abatement, while at higher frequencies, increased noise abatement is measured. Noise reductions of up to 11 dB for airfoils with 20% chord-length porous-extent at zero angle of attack was observed [30]. In a more recent study, Rubio Carpio et al. [16] employed planar particle image velocimetry (PIV) to quantify the flow field characteristics over the porous insert. They reported an increase in boundary layer and displacement thickness, a decrease in the mean velocity magnitude, and a reduction of velocity fluctuations. They attributed the low-frequency noise abatement to the role of the porous insert in changing the ratio of the wall-normal to the wall-parallel velocity components. There have also been a few analytical and numerical studies on the optimal distribution of rigid porous media using a discrete adjoint-based optimization framework [26,27].

In all previous studies, the porosity level is fixed by the manufacturing procedure. In this study, we propose the application of a polymeric coating to change the porosity of the open-cell structure, and as a result, its permeability level. This opens a new possibility of aeroacoustic control, since the polymeric coatings are temperature-responsive, and their dimension can be controlled inside the foam. In the present work, in order to analyze the coating response as a function of the permeability and internal foam architecture, four topologically comparable metallic foams of different average pore size are internally coated with a polymeric coating. Through a multiple-step coating process, a broad range of permeability values is obtained. The polymeric coating was selected for its good surface coverage behavior. The material is tested at room temperature and with multiple layers, in order to compare metallic and coated foams having the same pore size.

Extra attention is dedicated to verify that the rubbery material-state at room temperature does not alter in any manner the pressure fluctuation spectrum of the incoming flow. A modified aerosol spray-coating technique was used to obtain a uniform coverage of the internal foam structure. Flow resistivity and wind-tunnel tests at several free-stream flow velocities were performed to evaluate the permeability and aero-acoustical footprint of the coated and uncoated foams with respect to different Reynolds numbers.

## 2. Open-Foams with Polymer Coating

### 2.1. Foam Material

Four different open-cell Ni-Cr-Al foams with average cell diameters (*d_c_*) of 450 μm, 580 μm, 800 μm, and 1200 μm, have been employed in this study. An example of the coated 1200 μm foam is presented in two configurations (porosity characterization cylindrical sample and tested trailing edge insert) in Figure 1.

The metal foams are manufactured by the company Alantum (http://www.alantum.com, München, Germany). As reported by the company, the fabrication of the metal foam starts with the preparation of a sacrificial polymeric foam specimen, with the open-cell structure of the final product (dodecahedron cells). An electroplating technique is then applied to deposit Ni atoms on the surface to create a homogenous layer on the original specimen. Alloying elements are deposited afterwards with a spray method. The original polymeric scaffold is finally burned away while the alloy is being homogenized via a high temperature heat treatment [31]. The strut thickness in the foams results in 50 μm, with local variations. The uncoated foams tested in this study have porosities in the range of 89.3–91.7%. Porosity is defined as $p=1-\rho_{f}/\rho_{s}$ where $\rho_{f}$ and $\rho_{s}$ are respectively the densities of the foam and the solid material the foam is made of (Ni-Cr-Al).

### 2.2. Polymeric Coating Applied to the Open-Cell Foam

A dedicated spray methodology with a solvent-based paint Rust Oleum^®^ 2125 (Vernon Hills, IL, USA), is employed to obtain a uniform coating. The polymeric material was applied on metal foams with pore sizes of 580 µm, 800 µm, and 1200 µm, respectively. A 10-second aerograph deposition from a distance of ~2 cm from the sample surface, followed by a post treatment of the wet painted foam with a controlled air flow resulted in a uniform coating distribution within the foam samples (Figure 2). Up to six layers of paint with intermitted drying were applied to obtain desired combinations of the final pore size (i.e., flow resistivity) and coating thicknesses. Verification of the uniformity of the coating throughout the samples was carried out with 3D optical measurements and serial sectioning on both dedicated and tested samples (Keyence VHX 2000, Osaka, Japan). DMA measurements carried out using an RSA-G2 Solids Analyzer (TA Instruments, 159 Lukens Drive, New Castle, DE, USA) to obtain the dynamic mechanical properties of the coating material itself.

For such measurements, free standing rectangular films of 12 × 30 mm^2^ and 292 µm thickness were employed. The stress frequency was maintained at 1 Hz, and the maximum strain per cycle was 0.05%. The temperature was varied from −100 °C to 100 °C with rate of 3 °C/min.

### 2.3. Verification of the Coated Pore Size

Both porosity and density of all tested foams (coated and uncoated) were measured using Archimedes’ technique. All foams were cut into cubic specimens with approximate dimensions of 15 mm × 15 mm × 15 mm (giving 3375 mm^3^). The external volume of each sample was measured using a caliper. Measurements were performed to determine the effective volume of the interconnected hollow part, yielding the porosity of the four pore size categories. Archimedes’ measurements provide with the volume occupied by the solid part $V_{S}$ of the foam by the following equation:(1)$$V_{S}=\alpha \frac{M_{A}-M_{B}}{\rho_{l}-\rho_{air}}$$
where $M_{A}$ and $M_{B}$ are the specimen mass measured respectively in air and liquid, $\rho_{l}$ is the density of the liquid, $\rho_{air}$ is the density of the air, and *α* = 0.99985 is a calibration value. By knowing the total volume of the specimens $V_{T}$ (measured by a caliper), the percentage of total interconnected hollow space $V_{P\%}$ inside the porous structure can be obtained:(2)$$V_{P\%}\;=\;\frac{V_{T}-\;V_{S}}{V_{T}}$$

With a homogeneous paint layer throughout the structure, the average local paint thickness can be directly obtained by the measured increase in volume of the original foam (in this study further verified by microscope measurements). Since open-cell porous structures are composed of ligaments (almost one-dimensional elements), the volume of the void area inside the porous structure is proportional to the area of 2D pores in the facet of the porous structure. If the average diameter of the 2D pores in the facets of the porous structure initially measures D, the average diameter will decrease to d after applying the paint coating:(3)$$d=D\sqrt{1-\frac{\Delta V_{p}}{V_{p}}}$$
where $\Delta V_{p}$ is the decrease in the volume of hollow space in the porous structure due to the paint layer, and $V_{p}$ is the initial volume of the hollow space. The paint layer thickness t reads:(4)$$t=\frac{D-d}{2}=\frac{D}{2}\left(1-\sqrt{1-\frac{\Delta V_{p}}{V_{p}}}\right)$$

Multi-step coating applications treatments were imposed on test samples of foams with the larger pore sizes to determine those conditions which would lead to the same average pore size as for the next finer uncoated foam.

### 2.4. Air Flow Resistivity Tests

The static pressure drop was measured using a Mensor 2101 differential pressure sensor (Mensor Corporation, San Marcos, TX, USA), connected to the pressure taps with the apparatus shown in Figure 3a.

The pressure drop across the metal foams was measured between two pressure taps displaced at 5 cm distance upstream and downstream the specimen. For each pore size, the resistivity of two identical stacked coated or uncoated metal foam disks (Appendix A) with diameters of 55 mm and thicknesses of 10 mm (total thickness of 20 mm), was tested. The volumetric flow rate with an entry pressure of 10 bar was controlled using an Aventics pressure regulator. A TSI 4040 volumetric flow meter was used to measure the air flow upstream the main pipe. The permeability of the metal foam was then calculated using the Hazen-Dupuit-Darcy equation [32]:(5)$$\frac{\Delta p}{t}=\frac{\mu }{K}v_{d}+\rho Cv_{d}^{2}$$
where $\frac{\Delta p}{t}$ is the pressure drop normalized to the thickness t of the materials, μ is the dynamic viscosity, ρ is the fluid density, K is the permeability of the material which accounts for pressure loss due to viscous effects, and C is the form factor accounting for the pressure loss due to inertial effects. The parameter $v_{d}$ is the Darcian velocity which is defined as $v_{d}\;=\;Q/A$*,* where Q and A are the volumetric flow rate and the cross-section area of the sample, respectively. Following the definition from literature [28,29], the resistivity of the metal foam reads:(6)$$R\;=\;\frac{µ}{K}$$
with μ obtained using Sutherland’s law [33]:(7)$$\mu =\;\frac{C_{1}T^{\frac{3}{2}}}{T+S}$$
where T is the absolute temperature, and S = 110.4 K and $C_{1}=1.458\times 10^{-6}\frac{kg}{ms\sqrt{K}}$ for air. In this study with T=293.1 K, the dynamic viscosity results: $\mu \;=1.813\times 10^{-5}\frac{kg}{ms}$.

### 2.5. Aeroacoustic Measurements of the Porous Materials

Trailing-edge inserts with 10% chord length were designed and cut from foam panels. Coated and uncoated trailing edges were mounted on a NACA 0018 airfoil with a chord length of c = 0.2 m and span length of L = 0.4 m (span-chord ratio of L/c = 2), already employed in a previous experimental study using the metallic foams in an uncoated state [30]. The airfoil had an exchangeable trailing edge section which allowed for the testing of the airfoil with different porous trailing edge types. Measurements of sound pressure level from the trailing-edge inserts were performed in the anechoic vertical tunnel (AV-Tunnel) available at TU Delft. In the present configuration, the test section with a contraction ratio of 15:1 allows reaching free-stream velocities up to 42 m/s. The selected operated air stream velocities were 20, 30, and 40 m/s with a measured turbulence intensity of less than 0.1%. The NACA airfoil was mounted in a rectangular test-section (with dimension 40 × 70 cm^2^) and kept at 0° (details in Figure 3).

The sound signal was recorded by a planar microphone array holding 64 G.R.A.S. 40PH free-field microphones (Figure 3d). The microphones had a frequency response of ±1 dB, a frequency range of 10–20 kHz, maximum output of 135 dB and nominal phase spreading of ±3°, with CCP pre-amplifiers integrated. The microphones were arranged in an Underbrink multi-arm spiral configuration [34] with 7 arms and 9 microphones per arm, plus one in the center. The array effective diameter was 1.9 m. The array was placed at y = 1.43 m, according to the axes in Figure 3. The center of the array was approximately aligned with the center of the airfoil trailing edge. The turbulent boundary layer transition was forced at 20% chord length at both pressure and suction sides using carborundum particles (diameter of 840 μm) attached randomly on a tape strip.

The sampling rate for the pressure-time signal was 50 kHz, and each recording was carried out for 30 s. The sound pressure level (SPL), $L_{p}$, per 6.1 Hz frequency bandwidth was obtained by applying the Source Power Integration method for Line source (SPIL) [35]. The integration area is a rectangular grid with the width of 14 cm and the length of 27 cm chordwise and spanwise aligned with the airfoil principal axes, respectively. The simulated line source is assumed to be in the middle of the rectangle.

Although the location of the simulated line source is fixed, the source location for different frequencies may shift slightly from the location of the simulated line sources. This may cause small inaccuracies in the computed SPL [36]. The source map derived from beamforming is additionally subjected to the Rayleigh’s resolution limit [37] indicating the minimum source separation distance where different sound sources can be resolved using a finite-aperture array. The Rayleigh’s resolution limit (*RL*) is a function of the distance from the array to the scan plane (*y*), the speed of sound (*c*), the frequency (*f*), and the array’s effective diameter (*D*) and reads:(8)$$RL≅1.22y\;\frac{c}{fD}$$

## 3. Material Characterization and Aeroacoustic Results

### 3.1. Material Characterization

A first characterization of the coating procedure including the material deposition is carried out. Results from this first study show that it is indeed possible to change the material porosity through coating in a controlled way. In particular, increasing the number of deposition layers decreases the porosity of the coated porous material (Figure 4a). For the specimens with large pore sizes (i.e., 800 μm and 1200 μm), there was a direct proportionality between the porosity versus layer number, indicating a repeatable deposition procedure.

To calculate the thickness of the paint, Equation (5) was employed. As shown in Figure 4b, there is a modest change in the slope of the thickness versus number of coating steps curve at coating step 1, which can be attributed to the difference in the paint adherence characteristics with the bare metallic struts on one hand, and to struts coated in a previous deposition treatment on the other hand. The decrease in the slope of the curve with respect to the number of layers after the first layer of paint suggests that the metal surface provides a higher degree of adherence for the paint.

The combination of the right starting pore size and the right number of coating steps made it possible to create three sets of coated/uncoated samples with either similar pore size or permeability levels. As it can be seen in Figure 4c, the average pore size of the 1200 μm metal foam was decreased to 807 μm after applying 3 layers of coating (the results of which are to be compared to those of uncoated 800 µm metal foam). Moreover, 3 layers of coating over the metal foam with initial pore size of 800 μm led to final pore size of 562 μm (which is to be compared to 580 μm uncoated). On the other hand, for the case of the metal foam with an initial pore size of 580 μm, only one layer of coating was applied. The relevant process and geometrical parameters of the final set of three internally coated foams to be tested for permeability, and to be used in the wind tunnel experiments are listed in Table 1. The E’-T plot of the polymer coating material (Figure 5) showed the polymer to have a glass transition temperature of 44.5 °C, and a low temperature modulus in the glassy state of about 1.83 GPa.

At the nominal air temperature, the polymer dynamic modulus is about 43 MPa. For the conditions of the aeroacoustical test, the polymer coating is in a state where its mechanical excitations are (almost) optimally dampened.

### 3.2. Air Flow Resistivity Characterization

The results of the permeability measurements on the uncoated and coated material are shown in Figure 6 and are reported in Table 2. The results show that by internally coating the foams, about a two-fold increase in the resistivity can be achieved, while the permeability can be reduced by about 50% and the form coefficient can be increased by a factor of five, approximatively. The coated and uncoated foams cover partially overlapping permeability ranges and together span a permeability range of 0.497 to 5.612 m^2^. However, in none of the cases there was a perfect overlap of the air flow characteristics of the coated foam with the reference uncoated foams.

### 3.3. Far-Field Noise Measurements and Noise Reduction Validation

The differences between far-field noise reduction of uncoated porous trailing edges as compared to solid trailing edges are presented in Figure 7. The air velocity had a significant influence on the frequency range in which the implementing porous trailing edge was found to be effective (Figure 7). In particular, for the large pore size (i.e., 800 μm), the frequency at which noise reduction is observed simply shifts to higher frequencies with higher air velocities, but the shape of the curve and the magnitude of the maximal noise reduction is more or less maintained. For the lowest pore size (i.e., 450 μm) the onset frequency for noise reduction does not shift, but the level of reduction rises more gradually with increasing air velocity (Figure 7c). The behavior of the 580 μm foam (Figure 7b) forms a smooth transition between the velocity dependence for the foams with the largest and the smallest pore sizes.

Figure 8 aims to present a direct comparison between the noise abatement of the uncoated and the coated foams for the three air speeds explored. In each subfigure, the noise reduction for the initial (larger) pore size and the noise reduction for the uncoated foam are plotted with dashed lines, while the noise reduction for the coated foam is plotted as a solid line. The results show very clearly that foams with a comparable nominal pore size (for both the uncoated foam or the coated foam) yielded very comparable noise-reduction values, both in magnitude of damping and in frequency range over which this damping was observed. Only in the case of the foams with the smallest pore size (510 μm coated pore size and 450 μm uncoated) were there differences in the noise reduction spectra at lower frequencies, which are attributed to structure differences between the 450 μm foam and the foams with the larger pore size. In contrast, the correspondence between the noise spectra for the original 580 μm uncoated foam and the 510 μm coated foam, both having the same foam topology, was very high.

The frequency range for which the airfoil with a porous trailing edge (coated or uncoated) generates less noise than the airfoil with a solid trailing edge (i.e., $L_{p}$ > 0) is plotted in Figure 9, where the lower frequency limit of 300 Hz is set by the minimal frequency detectable as from the wind-tunnel anechoic characteristics. The maximum frequency at which noise reduction was recorded is highly velocity dependent (~2400 Hz for V = 20 m/s, ~3500 Hz for V = 30 m/s, and ~4500 Hz for V = 40 m/s, corresponding to *St* equal to 24, 23 and 22.5, respectively). Figure 8 and Figure 9 clearly demonstrate that the noise reduction behavior of coated and uncoated foams is indistinguishable, showing that the internal coating method is an excellent method for tuning the permeability of a foam, while keeping the noise reduction behavior equal.

Finally, while the presence of a compliant coating on a solid trailing edge has been reported [38] to have a positive effect on boundary layer thickness and the sound pressure levels generated, in the present experiments no real effect of the presence of the coating was observed, neither on the absolute noise reduction level, nor on the effective frequency range. Kulik et al. [39] suggested that in order to have energy dissipation due to viscous damping of the turbulent air structures, the resonant frequency of the internal contract/expansion of the coating material should be close to the frequency of the turbulent fluctuations.

The resonant frequency of a complaint material of thickness t can be approximated as [39]:(9)$$\omega_{0}=\frac{C_{t}}{t}\left(2.244+1.96\nu_{c}\right)$$
where $C_{t}$ is the velocity of the shear waves propagation and $\nu_{c}$ is the Poisson’s ratio of the coating material. $C_{t}$ equals:(10)$$C_{t}=\sqrt{\frac{E_{c}}{2\rho_{c}\left(1+\nu_{c}\right)}}$$
where $E_{c}$ and $\rho_{c}$ are respectively the elastic modulus and density of the coating material. Inserting appropriate values for the coating material used ($\rho_{c}=729\;\mathrm{kg}/m^{3}$, $E_{c}\approx 1\;\mathrm{GPa}$, and $\nu_{c}\approx 0.5$) yields a resonance frequency of around 1 MHz for the coating material, which implies that the coating material used indeed cannot be in resonance with the air flow fluctuations. To have such a dampening effect, a much softer coating with a modulus equal to that of a hydrogel (approximately 1 kPa) or a thicker coating should have been used, both of which solutions are unrealistic.

## 4. Conclusions

In this research, porous metallic foams with different initial average pore sizes ranging from 1200 μm to 450 μm were internally coated with elastomeric coatings of various thicknesses in order to adjust their air flow resistivity behavior, in particular their permeability, while keeping the topological complexity of the aerodynamic passage through the foams constant. Porous trailing inserts made out of these uncoated and coated foams were used as 10% chord trailing edge inserts in a NACA 0018 airfoil. Permeability tests in combination with aeroacoustic tests at zero angle of attack showed that the presence of the coating had a large effect on the air flow resistivity of the coatings, but had no real effect on the frequency dependence of the noise reduction, nor on the absolute noise reduction levels. This study opens the possibility of using the thermal characteristics of the polymer to control the porosity and therefore the noise reduction obtained by such porous materials when used at the edge of aerodynamic geometries.

## Figures and Tables

**Figure 1 materials-12-01087-f001:**
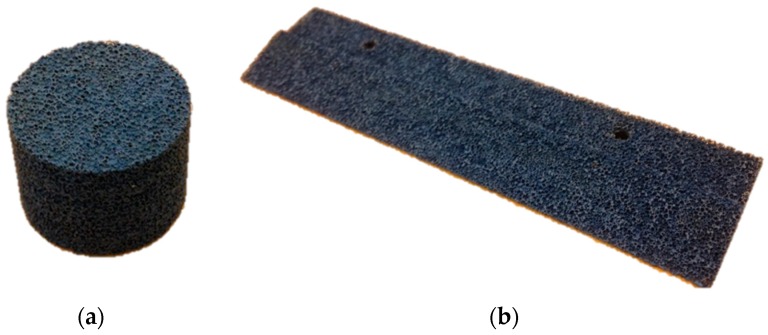
Examples of (**a**) cylinder of 2 cm diameter, and (**b**) trailing edge insert of 14 cm length manufactured by water-cutting of 1200 μm permeable foam.

**Figure 2 materials-12-01087-f002:**
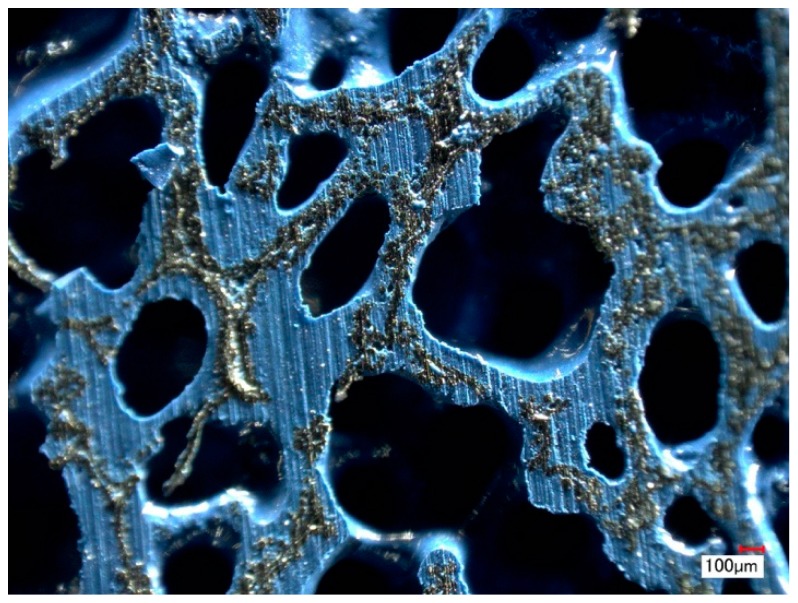
Cross-section of the trailing edge spray-coated by solvent-based paint.

**Figure 3 materials-12-01087-f003:**
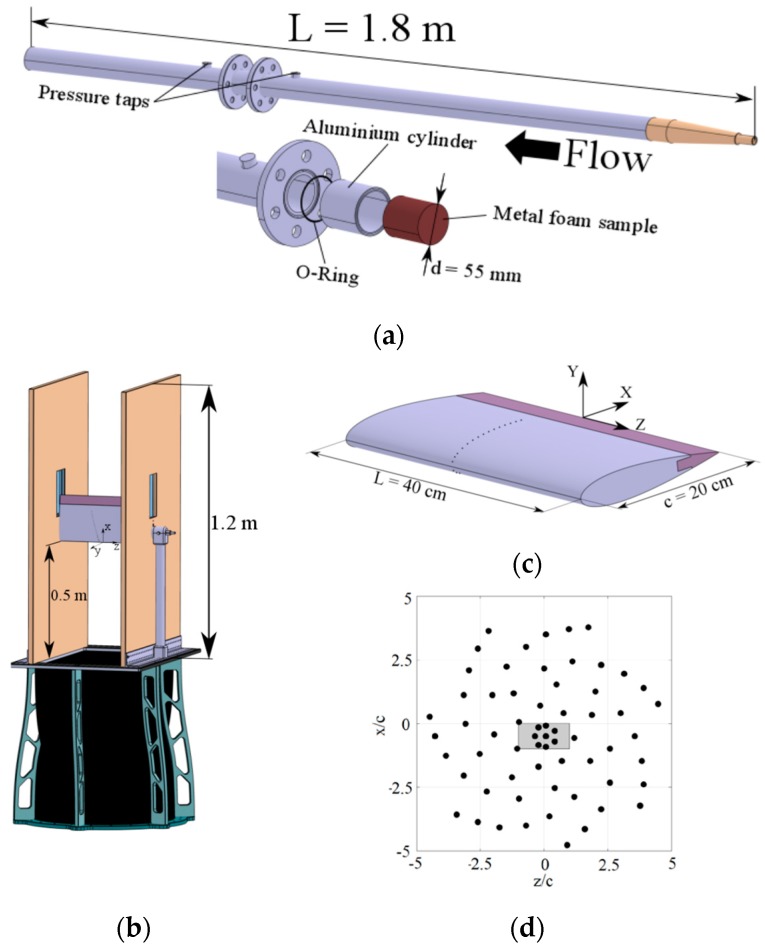
Setup of the (**a**) pressure measurement pipe for the permeability tests, (**b**) Wind tunnel schematics for the acoustic measurements, (**c**) the airfoil model dimensions, and (**d**) microphone array disposition (the grey part in the center represents the airfoil position with respect to the microphone array) drawings adapted from Rubio et al. [16].

**Figure 4 materials-12-01087-f004:**
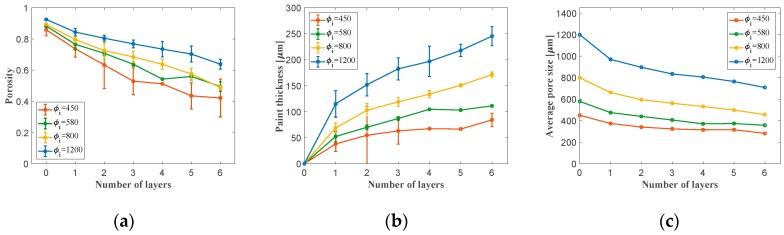
Variation of the (**a**) porosity, (**b**) paint thickness, and (**c**) average pore size with respect to the number of paint layers.

**Figure 5 materials-12-01087-f005:**
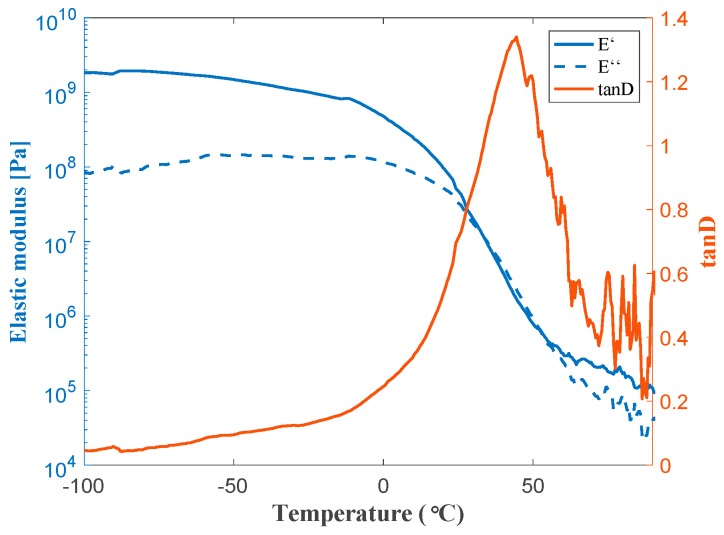
DMA measurements for solvent-based paint system.

**Figure 6 materials-12-01087-f006:**
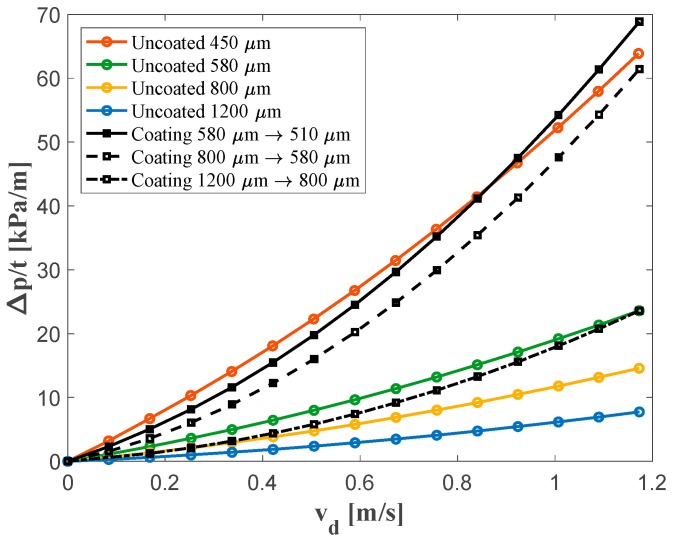
Comparison of measured pressure drop Δp along uncoated and coated metal foams normalized with respect to thickness t for different flow velocities $v_{d}$. Color coding as in Figure 4.

**Figure 7 materials-12-01087-f007:**
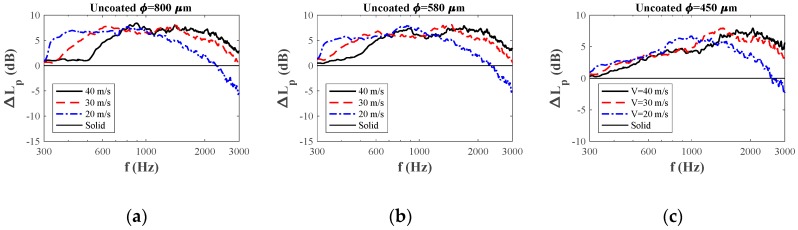
Comparison of noise abatement $\Delta L_{p}$ diagrams of uncoated trailing edges with (**a**) 800 μm, (**b**) 580 μm, and (**c**) 450 μm pore size.

**Figure 8 materials-12-01087-f008:**
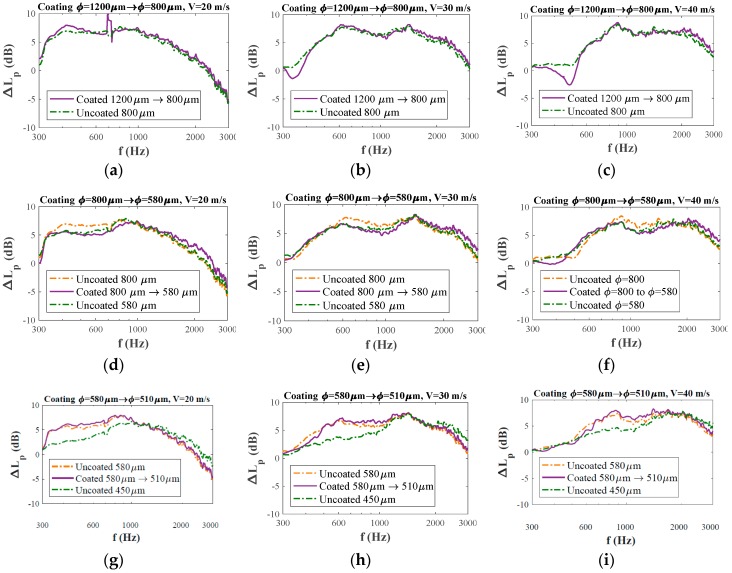
Comparison of noise abatement $\Delta L_{p}$ between coated and uncoated trailing edges at different frequencies f with respect to the solid reference edge for initial pore sizes of (**a**) 1200 μm in V = 20 m/s, (**b**) 1200 μm in V = 30 m/s, (**c**) 1200 μm in V = 40 m/s, (**d**) 800 μm in V = 20 m/s, (**e**) 800 μm in V = 30 m/s, (**f**) 800 μm in V = 40 m/s, (**g**) 580 μm in V = 20 m/s, (**h**) 580 μm in V = 30 m/s, and (**i**) 580 μm in V = 40 m/s.

**Figure 9 materials-12-01087-f009:**
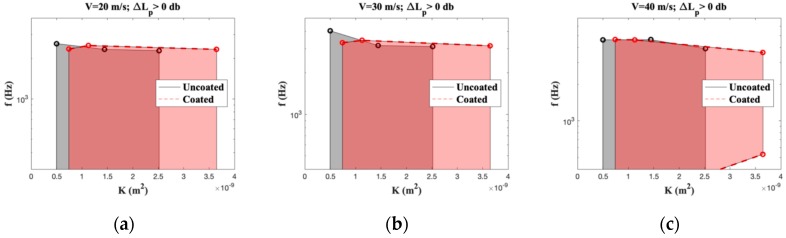
Comparison of frequency range f for which noise reduction is positive at different permeabilities K for different free-stream air velocities: (**a**) v = 20 m/s, (**b**) v = 30 m/s, and (**c**) v = 40 m/s.

**Table 1 materials-12-01087-t001:** Process conditions and geometrical parameters of the final set of internally coated foams used for aeroacoustic testing.

Initial Pore Size	Coating Steps	Final Pore Size	Coating Thickness	Reference Uncoated
[μm]	[n]	[μm]	[μm]	[μm]
1200	3	807	182.5	800
800	3	562	118.93	580
580	1	510	34.5	450

**Table 2 materials-12-01087-t002:** The measured resistivity, permeability, and form coefficient for uncoated and coated metal foams.

Sample	Resistivity, *R*	Permeability, *K*	Form Coefficient, *C*
	[Ns/m^4^]	[m^2^]	[1/m]
Uncoated 450 μm	36499	0.497 × 10^−9^	12,628
Uncoated 580 μm	12,585	1.441 × 10^−9^	5291
Uncoated 800 μm	7199	2.518 × 10^−9^	3678
Uncoated 1200 μm	3230	5.612 × 10^−9^	2366
Coated 580 μm→510 μm	24,465	0.741 × 10^−9^	24,019
Coated 800 μm→580 μm	16,053	1.129 × 10^−9^	25,494
Coated 1200 μm→800 μm	4969	3.648 × 10^−9^	10,614

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
