# Peer review of "Role of Polymeric Coating on Metallic Foams to Control the Aeroacoustic Noise Reduction of Airfoils with Permeable Trailing Edges"

_materials, 2019, doi:10.3390/ma12071087_

Round 1
Reviewer 1 Report
The article discusses role of polymeric coating on metallic foams to control the aeroacoustic noise. The article is undoubtedly interesting, but, unfortunately, it is necessary to correct it. There are a lot of typos.
line 11: Typo: Correspondence – twice
Abstract rather briefly and at the same time capaciously represents the main idea and conclusions of the article.
The introduction gives a fairly complete picture of research in this area, after analyzing the state of the problem, the authors formulate the research objectives in a rather focused manner.
Line 100: Error!
Line 118: Error!
Figure 2: the scale bar should be added.
Line 154, 187, 189, 216, 221, 230, 237, 249 .... : Error!
Figures 6 and 7: It is advisable to indicate not only the designation of the axes, but also the decoding. (what is delta p / t or vd?)
Table 2: Decoding of symbols is required (R, K, C), it is necessary to use a single style of decimal fractions ( dot or comma).
Figures 8 and 9: it is desirable to decipher the notation (f, delta L, k) - in the figure caption
Author Response
The article discusses role of polymeric coating on metallic foams to control the aeroacoustic noise. The article is undoubtedly interesting, but, unfortunately, it is necessary to correct it. There are a lot of typos.
1. line 11: Typo: Correspondence – twice
Response: It is now corrected.
2. Abstract rather briefly and at the same time capaciously represents the main idea and conclusions of the article.
Response: We would like to thank the reviewer for their compliment about the abstract.
3. The introduction gives a fairly complete picture of research in this area, after analyzing the state of the problem, the authors formulate the research objectives in a rather focused manner.
Response: In the last paragraph of the introduction, the following sentences are now minorly changed to formulate the research objectives in a more focused manner:
“In all previous studies, the porosity level is fixed by the manufacturing procedure. In this study, we propose the application of a polymeric coating to change the porosity of the open cell structure and as a result its permeability level. This opens a new possibility of aeroacoustic control, since the polymeric coatings are temperature responsive and their dimension can be controlled inside the foam.”
4. Line 100: Error!
Response: It is now corrected.
5. Line 118: Error!
Response: It is now corrected.
6. Figure 2: the scale bar should be added.
Response: The scale bar is now added.
7. Line 154, 187, 189, 216, 221, 230, 237, 249 ....: Error!
Response: It is now corrected.
8. Figures 6 and 7: It is advisable to indicate not only the designation of the axes, but also the decoding. (what is delta p / t or vd?)
Response: The designation of the axes is now added to the captions of Figures 6, 7, and 8.
9. Table 2: Decoding of symbols is required (R, K, C), it is necessary to use a single style of decimal fractions (dot or comma).
Response: The symbols are now explained in the first row of the table.
In order to not make mistake between decimal point and the 1000 comma, the commas are now removed from the data in the table.
10. Figures 8 and 9: it is desirable to decipher the notation (f, delta L, k) - in the figure caption
Response: The notation of f, delta L, and K is now added to the caption of Figures 8 and 9.
Reviewer 2 Report
In this paper, the effect of polymer coating on metallic foams to control the aeroacoustic noise reduction of airfoils was presented. The paper is clearly written and presented, however more important information should be added and clearly stated, so the article could be published after some modifications. My comments and concerns are listed below:
1. There are many “Error! Reference source not found..” throughout the manuscript. Please revise this section.
2. In figure 8, the velocity of the figure presented in the last row and second column should be V= 30 m/s instead of V=20m/s?
3. Since there is no significant difference of the noise reduction with the coated and uncoated foams, can authors give suggestion for reducing the noise in relating to the permeability or pore size of the foam?
Author Response
In this paper, the effect of polymer coating on metallic foams to control the aeroacoustic noise reduction of airfoils was presented. The paper is clearly written and presented, however more important information should be added and clearly stated, so the article could be published after some modifications. My comments and concerns are listed below:
1. There are many “Error! Reference source not found.” throughout the manuscript. Please revise this section.
Response: It is now corrected.
2. In figure 8, the velocity of the figure presented in the last row and second column should be V= 30 m/s instead of V=20m/s?
Response: Many thanks for your attention. The figure is now updated.
3. Since there is no significant difference of the noise reduction with the coated and uncoated foams, can authors give suggestion for reducing the noise in relating to the permeability or pore size of the foam?
Response: Rubio Carpio et al. (2019)* have shown that by increasing the permeability, the low frequency noise reduction increases for up to 5 dB but, at the same time, at high frequency, roughness induced noise is larger (up to 4 dB).
*Carpio, Alejandro Rubio, Roberto Merino Martínez, Francesco Avallone, Daniele Ragni, Mirjam Snellen, and Sybrand van der Zwaag. "Experimental characterization of the turbulent boundary layer over a porous trailing edge for noise abatement." Journal of Sound and Vibration 443 (2019): 537-558.